# The Current Status of Breastfeeding Knowledge and Its Influencing Factors in Nursing Undergraduates: A Cross-Sectional Study in China

**DOI:** 10.3390/ijerph20010103

**Published:** 2022-12-21

**Authors:** Keqin Liu, Jinjin Guo, Weixi Deng, Yanwei Su

**Affiliations:** School of Nursing, Tongji Medical College, Huazhong University of Science and Technology, Wuhan 430030, China

**Keywords:** breastfeeding, nursing undergraduates, nursing students, cross-section study

## Abstract

The aim of this study was to explore the breastfeeding knowledge of nursing undergraduates and the influencing factors. Human milk (HM) is one of the most effective nutritional supplies to improve early development and physical health, but the current status of breastfeeding in China is still not optimal. The breastfeeding knowledge of perinatal women influences their feeding beliefs and behavior. Nursing undergraduates, as core professionals who will care for perinatal women and provide feeding guidance in the future, can significantly affect feeding behavior of mothers and their babies, so their knowledge of breastfeeding may have a potential impact on breastfeeding in China. However, studies on breastfeeding knowledge among nursing undergraduates in China are limited. A convenience sampling method was conducted in four medical universities in China, and eligible nursing undergraduates were selected. An online survey was collected from 5 July 2022 to 5 August 2022. Categorical data were reported as number and percentage, while continuous data were reported as mean ± SD. Multivariate linear regression was used to evaluate the association between influencing factors and breastfeeding knowledge. The overall mean score of the 460 returned questionnaires was 43.991 out of 100. The pass rate of the questionnaire was only 23.04%. Nursing undergraduates had a relatively better grasp of the benefits of breastfeeding and related advice (correct rates: 67.83%). Birthplace, only child or not, the course in obstetrics and gynecological nursing, the course in pediatrics nursing, and placements in maternity or neonatology units were relevant factors for breastfeeding knowledge (*p* < 0.05). Nursing undergraduates showed unsatisfactory breastfeeding knowledge. It is urgent to raise the knowledge level of breastfeeding among nursing undergraduates. Medical colleges should optimally structure a curriculum of breastfeeding knowledge. Furthermore, it is also necessary to improve the public’s understanding of breastfeeding and the whole society’s attention to breastfeeding in China.

## 1. Introduction

Human milk (HM) is the optimal nutrition source for infants [1]. International organizations, such as the World Health Organization (WHO) [2] and the United Nations Children’s Fund (UNICEF) [3], launched the “Baby-Friendly” hospitals initiative and recommended providing exclusive breastfeeding for the first six months. Breastfeeding, due to its important protective effects for mothers and newborns, has been emphasized worldwide.

Maternal benefits of breastfeeding include decreased risk of type 2 diabetes mellitus, breast cancer and ovarian cancer, etc. [4]. Infants who are breastfed also have a decreased risk of diarrhea, necrotizing enterocolitis, late-onset sepsis, and related complications [5]. However, despite all the benefits of breastfeeding, the status of the breastfeeding rate in China is still not optimistic. According to relevant research, the exclusive breastfeeding rate for infants under six months of age in China is only 29%, far below the world average of 43% [6]. The causes for unobjective breastfeeding rates are multifactorial, and the most critical factor is the knowledge that perinatal women have about breastfeeding, which could directly affect their breastfeeding beliefs and behaviors.

In recent years, scholars and policymakers have increasingly recognized that it is essential to improve breastfeeding status. In China, the Chinese government has launched the National Program of Action for Child Development (2011–2020) [7] and the Breastfeeding Promotion Action Plan (2021–2025) [8], both of which have stated the goal of achieving an “exclusive breastfeeding” rate of 50% and over in the first six months of life. However, the status of breastfeeding has not met the expected situation, which may be related to the fact that clinical nurses do not play an obvious positive role in guiding breastfeeding knowledge [9].

Clinical nurses are the direct caregivers of perinatal mothers, and nursing undergraduates are the central core reserve of clinical nurses. Nursing undergraduates, the largest and most trusted healthcare professionals of the future, can take on the responsibility of translating the most scientific knowledge about breastfeeding into practice and improving the current situation of breastfeeding in China. Previous studies have focused on healthcare providers and postnatal mothers’ knowledge regarding breastfeeding [10,11], but there is little research on the knowledge of nursing undergraduates about breastfeeding and its influencing factors.

The aim of this study was to investigate the status of breastfeeding knowledge among nursing undergraduates and to analyze the relevant influencing factors. Understanding these factors might facilitate nursing undergraduates in valuing breastfeeding knowledge, and it might encourage medical colleges to optimize the structure of a breastfeeding-related curriculum, so as to further improve the breastfeeding rate in China.

## 2. Materials and Methods

### 2.1. Sample and Data Collection

The descriptive cross-sectional survey was conducted through an online survey platform Questionnaire Star (https://www.wjx.cn, accessed on 1 July 2022) from 5 July 2022 to 5 August 2022. Four colleges were selected by a convenience sampling technique. The participants consisted of nursing undergraduates in the third or fourth year of nursing. The research team members contacted each university’s administrators and explained the study’s purpose and procedures. After obtaining approval from the relevant head of the School of Nursing at each university, a Quick Response (QR) code was sent to all selected colleges. Participants who were interested in this study could scan the QR code for the student group to assess the online link for the survey. When opening the survey link, it was necessary to read the purpose of the study and explicitly agree to participate. To ensure that participants did not engage repeatedly, there was a limit, in that the questionnaire’s link could only be opened once on the same mobile device. All valid questionnaire answers were automatically loaded into a data file and checked by two reviewers independently. A total of 460 nursing undergraduates were surveyed.

The online questionnaires were divided into two parts. The demographic part included gender, age, educational level, birthplace, whether an only child or not, family income, marital status, participation in breastfeeding training, course in obstetrics and gynecological nursing, course in pediatrics nursing, and placement in maternity or neonatology units. The breastfeeding-related part tested information on the following: (i) breastfeeding advice, (ii) breastfeeding contraindications, (iii) breastfeeding complications, (iv) breastfeeding strategies, (v) breastfeeding advantages, and (vi) mechanism of HM production.

### 2.2. Measures

The Nursing Student’s Knowledge about Breastfeeding Questionnaire was developed by Natividad Lopez-Peña et al. [12] at Universität Jaume I. The Chinese version was translated and revised by Dai et al. [13]. The authors obtained authorization by email. Breastfeeding knowledge was evaluated by the breastfeeding questionnaire, which comprised 14 multiple choice questions with four options, of which only one was true. One point was given for each correct answer and 0 points for each wrong answer, or if a question was not answered. The scores of all items were summed up as the total score, and weighted according to 100 points to get the standard score. The higher the total score of the questionnaire, the better the breastfeeding knowledge. This instrument was validated in a Chinese sample and presented suitable internal consistency (Cronbach’s α = 0.763) and test–retest reliability (CCI = 0.813).

### 2.3. Data Analysis

All data were analyzed by using IBM SPSS 26.0 (IBM Corp, Armonk, NY, USA). In descriptive data, categorical variables were represented by number and percentage. Kruskal–Wallis Chi (χ^2^) was used for comparison between groups. Continuous variables were represented by means ± SD. The Mann–Whitney U was used for comparison between groups. Finally, a multiple linear regression analysis was carried out to explore which explanatory variables were associated with the Questionnaire score. A *p* < 0.05 was statistically significant.

### 2.4. Ethical Considerations

This study was approved by the university’s ethical committee of the participating universities (Approval No. S1191). Informed consent was obtained from every nursing student involved in this study. The survey was anonymous and no identifiable information was obtained, and questionnaires were completed voluntarily.

## 3. Result

### 3.1. Demographic Characteristics

This study collected 460 questionnaires with an overall questionnaire recovery rate of 95.5% (Huazhong University of science and technology = 22.8%; 105; Zhengzhou University = 29.4%; 135; Chongqing Medical University = 23.9%; 110; Zunyi Medical University = 23.9%; 110). Participants’ socio-demographic and breastfeeding-related information are presented in Table 1.

The participants were mainly composed of students aged 18–25 (96.5%), with a majority of senior students (76.3%). The participants were 81.7% (376) female. Half of the participants were living in urban areas (52.2%) and 34.6% reported their family income at an average level. Regarding previous breastfeeding knowledge, about two-thirds (63.9%) were only children, and the majority had no HM knowledge training (83.5%). In the systematic study of breastfeeding knowledge, most participants had studied obstetrics and gynecology nursing (81.3%) or pediatric nursing (81.7%). Regarding placements in maternity or neonatology units, about one third were not experienced (38.0% and 39.6%).

### 3.2. Description of the Breastfeeding Questionnaire Results per Item

From the questionnaire results, a mean overall score of 43.99 out of 100 was obtained. The overall questionnaire pass rate was 23.04% (scores ≥ 60 were considered qualified). Item 2 (advice about artificial feeding) had the highest score (67.83%) and Item 3 (about breast hygiene) had the lowest score (23.26%). The correct rate of the fourteen breastfeeding knowledge items is shown in Table 2.

### 3.3. Factors Associated with Breastfeeding Knowledge of Nursing Undergraduates

As can be seen in Table 3, the results showed that the participants who lived in urban areas achieved significantly higher scores, compared to those participants who lived in rural areas (*p* = 0.015). The results indicated a significant difference in the overall score between participants who were an only child and those who were not, with only child participants scoring relatively higher (*p* < 0.05). Nursing undergraduates who received systematic breastfeeding courses had significantly higher mean scores (48.09 ± 28.06 and 47.64 ± 20.00) in the total questionnaire over those who have not studied such a course (*p* < 0.001). There was a positive correlation between internship in obstetrics or pediatrics and mastery of breastfeeding knowledge (*p* < 0.001).

### 3.4. Multiple Linear Regression Analysis

Multiple linear regression was conducted to examine factors associated with the breastfeeding knowledge of nursing undergraduates. As shown in Table 4, the breastfeeding knowledge scores of nursing undergraduates who lived in urban areas were higher than that of the nursing undergraduates who lived in rural areas (*p* = 0.013), and the nursing undergraduates who were an only child (*p* = 0.043) presented higher breastfeeding knowledge scores. Compared with nursing undergraduates who had not studied a course in obstetrics and gynecological nursing or pediatric nursing, those participants who studied a systematic breastfeeding course showed higher breastfeeding knowledge scores (*p* < 0.001). In clinical placements, nursing undergraduates who had experienced internships showed higher breastfeeding scores relative to those who had not had a placement in obstetrics or neonatology units (*p* < 0.001).

## 4. Discussion

### 4.1. A Limited Breastfeeding Knowledge Level among Nursing Undergraduates

Nursing undergraduates’ breastfeeding knowledge determines whether they have sufficient expertise in optimal feeding methods to provide professional practical guidance for perinatal women [14]. As reported in this study, the pass rate of the questionnaire was only 23.04%. The results of our study were similar to those of the study by Huang et al. in 2018 [15], which had a mean pass rate of 51.75%. This result indicated a status of insufficient breastfeeding knowledge among nursing undergraduates. Although the Chinese government has offered much propaganda and policy guidance in recent years, there is still no significant improvement in the area of breastfeeding and, in fact, there is a decreasing trend in pertinent knowledge and practice of breastfeeding in China. The lack of breastfeeding knowledge of nursing undergraduates leads to an inability to correctly guide and provide breastfeeding support for perinatal women, which affects the beliefs and behaviors of maternal breastfeeding [16].

The questionnaire mainly included six aspects of breastfeeding knowledge: breastfeeding advice, breastfeeding contraindications, breastfeeding complications, breastfeeding strategies, breastfeeding advantages, and the mechanism of HM production. The item with the lowest mean score was “artificial feeding advice”, indicating that nursing undergraduates were not able to give relevant breastfeeding advice correctly. Meanwhile, the item with the lowest correct rate was “breast hygiene” (23.26%). This result might be explained by a gap between theoretical knowledge and practical education on breastfeeding among nursing undergraduates. We found nursing undergraduates lacked adequate knowledge of breastfeeding, which was in line with previous studies [17,18]. Given the lack of systematic and comprehensive knowledge of breastfeeding, medical colleges should optimize the breastfeeding curriculum and enhance understanding of breastfeeding among nursing undergraduates.

### 4.2. Associated Factors Related to Breastfeeding Knowledge among Nursing Undergraduates

Urban area participants primarily formed our study sample. This is a common finding in this kind of study, as these studies are mainly conducted in universities. We found that more than half of nursing undergraduates living in urban areas had better knowledge of breastfeeding, which could be related to diversification of access to more breastfeeding knowledge in urban areas [19]. The majority of the participants who were an only child was striking. Meanwhile, most participants were unmarried nursing undergraduates with no experience of breastfeeding. This result indicated that although the surveyed only child students had no experience in breastfeeding, they showed great interest in perinatal care and breastfeeding in particular [20].

Yang et al. [15] concluded that theoretical and clinical education could improve breastfeeding knowledge among nursing undergraduates. In our study, nursing undergraduates who received systematic education related to breastfeeding scored higher, confirming that a well-developed curriculum could improve breastfeeding knowledge. More importantly, nursing curriculum should not only focus on coverage of breastfeeding, but also pay more attention to knowledge about the content and practice of breastfeeding, which significantly affects the level of knowledge learning and acquisition. Meanwhile, this study also found that nursing undergraduates on placements in maternity or neonatology units appeared to have higher breastfeeding knowledge. These findings presented consistency with the validity of the results of this study, which may be related to the following reasons. Firstly, differences in curriculum between different grades. In the medical colleges participating in our study, breastfeeding-related courses started in the junior year, mainly majoring in obstetrics and gynecology nursing or pediatric nursing. Secondly, integrating breastfeeding theory with clinical practice. The third- and fourth-year nursing undergraduates experienced clinical exposure and work in maternity or pediatric wards. This learning model allows nursing undergraduates to have a positive experience during clinical placements, as the clinical sites provide sufficient opportunities to achieve the skills needed to support women in learning about breastfeeding. In addition, environment-specific learning models can stimulate interest in learning about breastfeeding. A previous study reported that nursing students believe promoting breastfeeding is essential for infants’ physical health and quality of life [21]. Through clinical placement in hospital, nursing undergraduates had opportunities to observe and provide breastfeeding support to new mothers, develop their interest in breastfeeding, and promote breastfeeding practice.

## 5. Limitation

This study has several limitations. First, since this was a cross-sectional study, causal relationships between breastfeeding knowledge and the influencing factors could not be determined. Second, a convenient sampling method was used and only four medical colleges were covered in China, which may limit the transferability of these findings. Therefore, future studies are recommended to expand the study population and explore other influencing factors to reinforce our findings or to improve the results.

## 6. Conclusions

In our study, we found an overall lack of breastfeeding knowledge among nursing undergraduates in China. The influencing factors were birthplace, being an only child or not, attending a course in obstetrics and gynecological nursing, attending a course in pediatrics nursing, and experiencing placements in maternity or neonatology units. In the future, nursing undergraduates should be aware of their own lack of breastfeeding knowledge, so as to strengthen and focus on breastfeeding-related knowledge. Medical colleges need to reinforce the link between theoretical and practical education in breastfeeding. Furthermore, nursing undergraduates’ low level of breastfeeding knowledge also reflects the whole society’s lack of cognition of breastfeeding. In the future, it is also essential for the government to further vigorously advocate the popularization of breastfeeding knowledge in China.

## Figures and Tables

**Table 1 ijerph-20-00103-t001:** Characteristics of Participants (*n* = 460).

Variable	*n* (%)
Age	
<18	7 (1.5%)
18–25	444 (96.5%)
≥25	9 (2.0%)
Sex	
Male	84 (18.3%)
Female	376 (81.7%)
University	
Huazhong University of science and technology	105 (22.8%)
Zhengzhou University	135 (29.4%)
Chongqing Medical University	110 (23.9%)
Zunyi Medical University	110 (23.9%)
Current academic year	
Third	109 (23.7%)
Fourth and fifth	351 (76.3%)
Birthplace	
Urban	240 (52.2%)
Rural	220 (47.8%)
Marital status	
Spinsterhood	449 (97.6%)
Married	11 (2.4%)
Family income (thousand/month)	
<0.5	87 (18.9%)
0.5–1.0	159 (34.6%)
1.0–1.5	116 (25.2%)
1.5–2	47 (10.2%)
≥2	51 (11.1%)
Previous BF knowledge	
Only child or not	
Yes	294 (63.9%)
No	166 (36.1%)
Formal BF train	
Yes	76 (16.5%)
No	384 (83.5%)
Course in obstetrics and gynecology nursing	
Yes	374 (81.3%)
No	86 (18.7%)
Clinical placements in maternity units	
None	175 (38.0%)
<1 week	26 (5.7%)
1 week–2 week	49 (10.7%)
2 week–1 month	128 (27.8%)
≥1 month	82 (17.8%)
Course in pediatric nursing	
Yes	376 (81.7%)
No	84 (18.3%)
Clinical placements in neonatology units	
None	182 (39.6%)
<1 week	34 (7.4%)
1–2 week	40 (8.7%)
2 week–1 month	130 (28.3%)
≥1 month	74 (16.1%)

BF: breastfeeding.

**Table 2 ijerph-20-00103-t002:** Questionnaire item correct rate.

Questionnaire Item	Correct Rate
1 BF recommendations according to the WHO	48.62%
2 Artificial feeding advice	67.83%
3 Mastitis care	39.13%
4 Contraindications to breastfeeding	45.00%
5 BF frequency	44.35%
6 Two hours postpartum care	31.96%
7 Colostrum	58.70%
8 BF production	53.04%
9 Kangaroo care	39.13%
10 Benefits for breastfed infant	38.26%
11 Benefits for mother	57.39%
12 BF premature neonates	41.96%
13 Breast hygiene	23.26%
14 BF infant’s position	52.61%

BF: breastfeeding; WHO: World Health Organization.

**Table 3 ijerph-20-00103-t003:** The results of univariate analysis on the influencing factors of breastfeeding knowledge (*n* = 460).

Variables	M	SD	Test	*p*-Value
Age				
<18	39.80	15.35	5.346	0.069 ^a^
18–25	44.40	22.07		
≥25	37.76	21.06		
Sex				
Male	41.50	23.20	−1.171	0.242 ^b^
Female	44.55	21.80		
Birthplace				
Urban	46.43	22.23	−2.424	0.015 ^b^
Rural	41.33	21.63		
Marital status				
Spinsterhood	43.99	22.05	−3.332	0.740 ^b^
Married	44.16	23.87		
Formal BF train				
Yes	47.65	20.98	−1.365	0.172 ^b^
No	43.27	22.23		
Family income (thousand/month)				
<0.5	43.84	21.07	1.830	0.767 ^a^
0.5–1.0	45.15	22.36		
1.0–1.5	43.90	21.75		
1.5–2	43.16	18.84		
≥2	41.60	26.48		
Only child or not				
Yes	45.55	22.24	−2.177	0.029 ^b^
No	41.22	21.55		
Current academic year				
Third	40.76	23.61	−1.887	0.059 ^b^
Fourth and fifth	44.99	21.50		
Course in obstetrics and gynecology nursing				
Yes	48.09	20.06	−8.029	<0.001 ^b^
No	26.16	21.63		
Clinical placements in maternity units				
None	38.37	23.37	18.318	<0.001 ^a^
<1 week	44.78	22.91		
1 week–2 week	46.50	23.15		
2 week–1 month	47.10	20.15		
≥1 month	49.39	18.79		
Course in pediatric nursing				
Yes	47.64	20.00	−7.337	<0.001 ^b^
No	27.63	23.65		
Clinical placements in neonatology units				
None	38.30	23.40	28.349	<0.001 ^a^
<1 week	39.71	23.87		
1–2 week	43.07	21.27		
2 week–1 month	48.30	18.82		
≥1 month	51.25	20.01		

SD: standard deviation; ^a^ Kruskal–Wallis Chi2; ^b^ Mann–Whitney U.

**Table 4 ijerph-20-00103-t004:** Multivariate analysis of breastfeeding knowledge scores of nursing undergraduates (*n* = 460).

Variables	β	SE	Standardized Coefficients (Beta)	t	*p*-Value
Birthplace (Ref = rural)					
Urban	−5.097	2.048	−0.116	−2.489	0.013
Only child or not (Ref = No)					
Yes	−4.332	2.135	−0.094	−2.029	0.043
Course in obstetrics and gynecology nursing (Ref = No)					
Yes	−21.927	2.435	−0.388	−9.004	<0.001
Clinical placements in maternity units (Ref = None)					
≤1 week	6.413	4.559	0.067	1.407	0.160
1–2 weeks	8.134	3.506	0.114	2.320	0.021
2 weeks–1 month	8.731	2.523	0.177	3.461	0.001
≥1 month	11.023	2.903	0.191	3.797	<0.001
Course in pediatric nursing (Ref = No)					
Yes	−20.008	2.497	−0.351	−8.013	<0.001
Clinical placements in neonatology units (Ref = None)					
≤1 week	1.401	4.019	0.017	0.349	0.727
1–2 weeks	7.767	3.756	0.099	2.068	0.039
2 weeks–1 month	9.992	2.470	0.204	4.045	<0.001
≥1 month	12.950	2.966	0.216	4.367	<0.001

F = 15.332, *p* < 0.001, R^2^ was 0.169 and the adjusted R^2^ was 0.158, indicating that approximately 15.8% of breastfeeding knowledge scores could be explained by the model. SE: standard deviation.

## Data Availability

Data may be provided by the corresponding author upon reasonable request.

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
