# Peer review of "The Current Status of Breastfeeding Knowledge and Its Influencing Factors in Nursing Undergraduates: A Cross-Sectional Study in China"

_ijerph, 2022, doi:10.3390/ijerph20010103_

Round 1
Reviewer 1 Report
Overall the papers covers a very important topic related to public health and makes appropriate suggestions based on results to improve breastfeeding rates at national level. some minor changes suggested are as listed below.
1. Nursing undergraduates are urgently need to raise awareness on the importance of breastfeeding.-- sentence needs correction
2. most direct caregivers-- remove "most"
3. litter research-- "litte"
4. A total of 460 nursing undergraduates were surveyed.-- do authors know the total numebr of nursing graduates enrolled in there 4 universitites? This will help better understand the popupation studied.
5. sentance 242-- Furthermore, as nursing undergraduates who have learned breastfeeding knowledge, their low level of breastfeeding knowledge also reflects the current situation that the overall low level of society.-- simplify the statement
6. Minor correction suggested to add word "area" after the word "urban" when used in text.
7. sentance 205 - can authors clarify this conclusion they made: "This result indicated that although the surveyed students had no experience being involved in breastfeeding, they showed great interest in perinatal care and breastfeeding in particular. " Do authors state this bacuase most participants were recognized as "spinster"?
8. sentance 215- " Meanwhile, this study also found that nursing undergraduates on placements in maternity or neonatology units appeared to have higher breastfeeding knowledge."-- its essential to recognize the curriculum differnece in early years of training as compared to third and fourth year. Did authors find that the third and fourth year trainees reported clinical exposure and work in maternity wards as compared to nurses early in phase of traning?.
Reviewer 2 Report
Nice cross sectional study of investigated question - education in breastfeeding. Simple in nature, and important in giving directions what-to-do.
As it was simple to conduct, maybe is possible to spread the questionnaire to more than four universities, in future, to have stronger evidence for decision-makers ?
The title of the paper describes the subject - the knowledge of undergraduates on breastfeeding in China. This topic in not so original but it is important from the public health viewpoint. It is a cross sectional study and shows the current situation in education about breastfeeding in four universities in China. Maybe this investigation in future could be extended on a bigger number of universities to get wider view on the extent of such low level of knowledge. Methodology is appropriate for cross sectional study. From the discussion, appropriate conclusions are drawn, what could bring changes in educational program for nurses.
